# Detection of social anxiety using multiple simultaneous biosignals: A pilot study

**Christoph Tremmel**[1]*, **Nathan T. M. Huneke**[2,3], **Daniel Hobson**[1], **M. C. Schraefel**[1]

**1** University of Southampton, Electronics and Computer Science, Southampton, Hampshire, United Kingdom, **2** Clinical and Experimental Sciences, Faculty of Medicine, University of Southampton, Southampton, Hampshire, United Kingdom, **3** Institute for Life Sciences, University of Southampton, Hampshire, United Kingdom

* christoph.tremmel@southampton.ac.uk

**Data availability statement:** Data used in this study are available at 10.5258/SOTON/D3622 or https://eprints.soton.ac.uk/503817/.

## Abstract

Social Anxiety Disorder (SAD) is a prevalent and debilitating condition characterized by heightened anxiety and avoidance behaviors in social situations. Despite the availability of treatment options, remission rates for SAD remain low, highlighting the need for more effective interventions. To support the development of more effective therapies, a better understanding of the psychophysiological mechanisms underlying SAD is needed. This pilot study investigates whether anticipatory anxiety before a social interaction can be detected by multiple biosignals, with the aim of identifying potential biomarkers for SAD. Using a modified version of the Internet-based Stress Test for Social Anxiety Disorder, we measured physiological responses of 17 healthy volunteers, including heart rate, respiratory rate, electrodermal activity, head movements, and electroencephalography power across various frequencies. We found that anticipatory anxiety was associated with increased heart rate, respiratory rate, tonic EDA, and variance in head roll, alongside elevated theta, beta, and gamma power in EEG. Our results suggest that a combination of these biosignals may provide valuable insight into the psychophysiology of social anxiety, which could be useful for both mechanistic research and clinical applications. Future research should explore the role of these signals in clinical populations and during acute threat conditions to refine diagnostic and prognostic tools for SAD.

## Introduction

Social anxiety disorder (SAD) is a chronic mental health condition defined by severe and enduring avoidance of performance or social situations, due to anxiety about perceived scrutiny by others. SAD is one of the most common anxiety disorders, with a lifetime prevalence of approximately 6-13% [1,2]. The peak onset is in adolescence/early adulthood [3–5], and it is highly comorbid with other mental disorders, substance misuse, and physical illnesses, leading to considerable societal costs. In 2010, anxiety disorders cost the UK an estimated £11 billion [6]. To our knowledge, this was the most recent formal estimate, but it is likely that current costs are even higher. Quantifying these economic burdens highlights the significant impact of anxiety disorders on public health systems and underscores the urgency

**Funding:** The authors acknowledge the University of Southampton's Researcher Development Concordat team for funding consumables in support of this study; the Web Science Institute Southampton that provided the funds for the experiment and equipment as part of a interdisciplinary pilot project scheme. This research is also supported by UKRI/EPSRC Grants EP/T007656/1: Health Resilience Interactive Technologies and EP/R029563/1: AutoTrust. The funders had no role in study design, data collection and analysis, decision to publish, or preparation of the manuscript.

**Competing interests:** There are no competing interests.

of improving prevention and treatment strategies. Current first-line treatments include cognitive behavioural therapy (CBT) and pharmacological treatments such as selective serotonin reuptake inhibitors [7]. However, almost half of patients with SAD do not achieve remission after four years, the lowest remission rate of all the anxiety disorders [8–10], indicating a need for more effective treatments. Despite this clear need, no new medications have been approved for any anxiety disorders since 2014 [11]. To develop new treatments, a deeper understanding of the mechanisms underlying anxiety and avoidance in SAD is essential. Higher level Biomedical measures may help identify new therapeutic targets and guide the development of more effective interventions.

Patients with SAD often focus their attention internally during a social interaction, overly scrutinising their physiological responses (e.g. increased heart rate (HR), breathing rate, flushing, muscle tension [12,13]). This information is used to build a negative visual image of the self that increases pre-interaction fears of negative evaluation and can trigger avoidance behaviours. Recently researchers have suggested that the interaction between interoception and physiological reactivity could provide a therapeutic target for development of new treatments [14]. Given these findings, physiological reactivity and interoceptive awareness might interact to maintain anxiety. For example, a patient with SAD might perceive their heartbeat as audible to others and believe they are being judged for this, resulting in avoidance behaviour. However, before this interaction between interoception and physiological reactivity can be targeted for treatment development, more understanding regarding how patients' physiology reacts to social anxiety is needed.

Converging evidence shows that SAD is associated with autonomic system dysfunction. This has been measured via reduced heart rate variability (HRV) [15,16] and blunted pupil dilation to faces [17,18]. But, these responses are non-specific to SAD and are also seen in generalised anxiety disorder, panic disorder, and post-traumatic stress disorder [19]. Similarly, previous research has shown that electrodermal activity (EDA), also called galvanic skin response (GSR), is associated with stress [20,21], but to our knowledge there are no results specific to *social anxiety*. Finally, increased theta power on electroencephalography (EEG) might be associated with social exclusion [22], however, other studies show that lower theta power is associated with social evaluation [23]. Instead, the *profile* of features taken together across modalities is likely to be more specific than a single modality that overgeneralises. Supporting this, anxiety during a driving hazard perception test is detected with higher accuracy when combining EEG, EDA and pupil size compared with single modalities [24]. There is growing interest in combining multiple biosignals to accurately measure cognitive and emotional states [25], but this has not been done in SAD. We carried out a pilot study to explore whether:

1. It was possible to detect anxiety in anticipation of a social interaction (even at a low level) by combining multiple biosignals and
2. If so, which of those signals are altered by anticipatory anxiety and therefore should be the focus for future studies in this field.

Here, we present preliminary results in which we have identified multiple physiological features that differentiate between low and high anxiety in the context of an impending social threat. In providing this preliminary information, it is our goal to help the field focus efforts on the biosignals that are likely to have the most value for mechanistic research and/or clinical utility for patients with SAD.

## Materials and methods

Participants were recruited from staff and students at the University of Southampton and provided written informed consent in accordance with the University of Southampton's ethical guidelines and the Declaration of Helsinki. Ethical approval was granted by the University's ethics committee in December 2023 (ERGO Application ID: 89571). Recruitment took place between 1.4.2024 and 1.7.2024. All participants were healthy individuals; those with a previous diagnosis of anxiety disorder were excluded. Each experimental session lasted roughly 90 minutes including preparation time. Eighteen participants (14 Male, Mean Age 34.8 STD 9.9) enrolled in this study and each received 20 GBP in compensation.

### Experimental procedure

Participants were sat comfortably while connected to 27-electrode EEG, 2-electrode electrocardiogram (ECG), 2-electrode galvanic skin response (GSR) and 4-electrode electrooculogram (EOG) using BrainProducts' "actiChamp plus" system. We used a MindMedia's "Nexus" respiration sensor with a custom connector to record respiratory rate, Pupil Lab's "Neon" to track eye movements, pupil size, head acceleration, angular velocity and orientation, Blue Microphones' "Blue Yeti" to record audio and used an ELP webcam model USBFHD01M-SFV to record video. EEG, ECG, EOG, GSR and respiratory rate were sampled at 500 Hz, eye movements and pupil size at 200 Hz, head acceleration, angular velocity and orientation at 110 Hz, audio at 48 kHz, and video at 30 Hz. All signals were synchronized using LabStreamingLayer [26]. We do not describe video and audio further in the present paper since this analysis focuses on biosignals.

As part of the "ActiChamp" system, "ActiCap" slim wet electrodes were placed at positions Fp1, F7, Fz, F3, FC5, T7, C3, Cz, TP9, CP1, P7, CP5, P3, O1, FP2, F8, F4, FC6, T8, C4, CP2, TP10, P8, CP6, P4, Pz and O2 according to the International 10-20 system [27]. Two EOG electrodes were placed above (FP2) and below the right eye to capture the vertical eye movement signal as well as two near the canthus of each eye for the horizontal signal. Two electrodes were placed at the insides of the elbow to get the ECG signal. Two GSR electrodes were placed on index and middle finger of the participants' non-dominant hand to get the GSR signal. All of the signals recorded by the actiChamp plus system underwent on-device analog high-pass filtering at 0.016 Hz. Good signal quality was ensured for each participant by visual inspection for all signals.

### Experimental task

The whole protocol is described in the supplementary materials section, however, for this paper only the final part was relevant to anxiety and is described further below.

Participants conducted an adapted version of the InterneT-based Stress Test for Social Anxiety Disorder (ITSSAD) by Huneke et al. [28]. Participants were told they would have five minutes to prepare for an online social interaction in which they would have to introduce themselves to a group of researchers. Simply informing participants about the upcoming interview has been shown to increase anxiety, so we did not conduct the full protocol for this pilot. For the exploratory purposes of this paper, we aimed to minimize any interference from artifacts potentially generated during the interview stage through conversation and facial expressions, as many biomedical measurements, especially EEG, are sensitive to such artifacts. After the five minutes had passed, participants were debriefed and informed that the task was designed to induce elevated levels of social anxiety without the actual interview. Then the experiment was finished.

During this task, anxiety was measured right before and right after the task through a modified version of the generalized anxiety disorder 7-item questionnaire (GAD 7, [29]), where each question was represented by a visual analogue scale as it can be seen in Fig 1. We have previously demonstrated that this modification allows measurement of changes in state anxiety with high sensitivity [28,30]. The first GAD7 was introduced before introducing participants to the anxiety task.

One participant was excluded from the subsequent analysis due to inconsistencies in their responses on the GAD-7, which raised concerns about the validity of their data. On review of the video footage, we found that this participant was not paying attention to the instructions on the screen, therefore the data was deemed unreliable for inclusion. Participants were then divided into two groups based on their responses to the GAD-7 questionnaire before and after the anxiety condition via a median split. This resulted in 8 participants in the non-anxious group and 9 in the anxious group.

## Data pre-processing and feature extraction

A large number of signals were recorded; therefore, the pre-processing and feature extraction section focuses on measures that either showed statistical significance for the anxiety condition or that were associated with increased anxiety in literature. All filters utilized in subsequent processing were zero-phase filters.

**Pre-processing.**

- Heart Rate & HRV: The HR signal was obtained by subtracting the ECG electrodes from one another and applying a band-pass filter between 2 and 30 Hz.
- Tonic GSR: The GSR signal was processed using a low-pass filter with a cutoff frequency of 0.05 Hz.

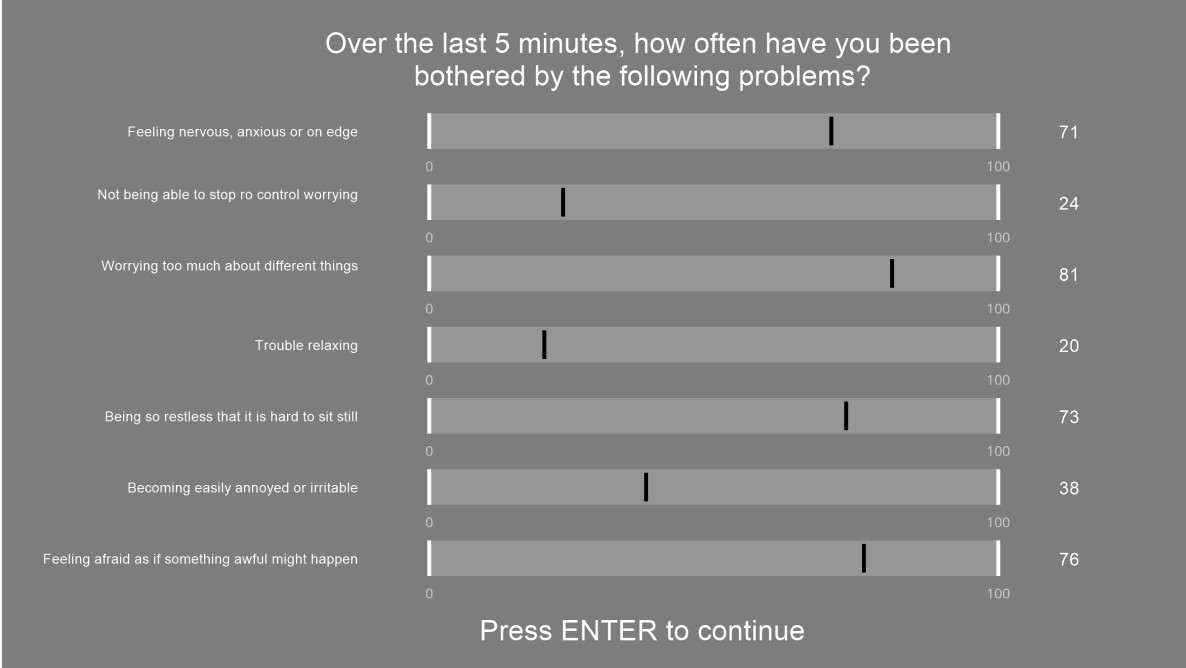

**Fig 1. The modified version of the generalized anxiety disorder 7-item questionnaire that was used during the study.**

- Phasic GSR: The GSR signal was processed using a high-pass filter with a cutoff frequency of 0.05 Hz.
- Pupil Size: The pupil size signal was filtered using a low-pass filter at 50 Hz.
- Respiratory Signal: A band-pass filter was applied to the respiratory signal with a frequency range of 0.1 to 0.5 Hz.
- Head Roll: The rolling motion of the head, recorded using the gyroscope, was low-pass filtered at 50 Hz.
- EEG: EEG channels were referenced to the average of electrodes located at TP9 and TP10, resulting in a total of 25 channels. EOG signals were utilized only for artifact removal. Vertical and horizontal EOG channels were derived by subtracting the two vertical EOG electrodes and the two horizontal EOG electrodes, respectively. Recorded signals were filtered between 0.5 and 50 Hz, with an additional 50 Hz notch filter applied to suppress power-line artifacts. Eye movement artifacts were suppressed in the resulting signals by creating a linear regression model to predict vertical and horizontal EOG from each EEG channel [31]. These predictions were then subtracted from the corresponding channels. The removal of EOG signals in EEG was applied to isolate and examine brain activity relevant to the task. Finally, a common average reference filter [32] was applied to EEG.

Since electrophysiological and gyroscope signals were recorded using different devices, precise synchronization had to be ensured. Instead of relying on the native timestamps from each device, we used the timestamps provided by LSL and resampled each of the signals to 100 Hz. The data were then segmented into epochs. Each epoch had a duration of 30 seconds, updated every 5 seconds, resulting in a total of 55 epochs.

**Feature extraction.**

- Heart Rate: HR was extracted using a peak detection algorithm applied to the normalized negative derivative of each epoch.
- HRV: HRV was extracted by calculating the standard deviation of all HR peak-to-peak intervals and dividing it by the root mean square of the peak-to-peak intervals.
- Tonic & Phasic GSR: The average tonic + phasic GSR was computed for each epoch.
- Pupil Size: The trace of the pupil size is corrected that the first epoch starts at zero. This was done to ensure that the distance between participant and experimenter for each condition was not influencing the outcome of the analysis. Each epoch was then featured by its mean.
- Respiratory Rate: A peak detection algorithm was used to extract the respiratory rate from the normalized respiratory signal.
- Head Roll: The roll signal was featured by its variance. Additionally, the power spectral density (PSD) was computed using Welch's method with 1 Hz frequency bins.
- EEG: The PSD for EEG data was also computed using Welch's method for each EEG electrode with 1 Hz frequency bins.

**Statistical analysis.** Participants were divided into "anxious" and "non-anxious" groups using a median split based on the change in self-reported anxiety from the pre- to post-anticipation phase. Individuals with change scores above the median were categorized as anxious, and those below as non-anxious. This method was chosen to create two approximately equal groups for balanced exploratory analysis in this pilot study.

We analyzed each biosignal modality separately. The figures in the results section that display the biosignals follow a consistent format. The left panel shows the time course of the relevant physiological measure from 30 to 300 seconds. Yellow shaded areas indicate time intervals where statistically significant differences between the two groups were identified

using a Wilcoxon rank-sum test. Due to the limited sample size in this pilot study, no correction for multiple comparisons was applied to the time series data. These results are intended to indicate where differences may occur and should be interpreted as exploratory, serving to identify potential time windows of interest for future, larger-scale studies. The right panel of each figure presents a boxplot summarizing the average of the analyzed measure across the full time window. It also includes the p-value for the difference between the groups over this interval, calculated using the same Wilcoxon rank-sum test. For the PSD analysis, significant frequency band differences were found for head roll and EEG data. P-values were displayed uncorrected and corrected using the Benjamini-Hochberg procedure.

## Results

The boxplots in Fig 2 show the differences in GAD-7 scores, averaged across all questions, for anxious and non-anxious participants from after to before the anxiety condition. Participants whose score differences were below the median were classified as non-anxious, while those above the median were classified as anxious. Table 1 summarizes the descriptive and inferential statistics for each measure and differences between groups. We further explore the results for each measure in turn below.

Fig 3 shows the heart rate during the anxiety task. On average, participants in the anxious group have a heart rate that is significantly higher than those in the non-anxious group. At

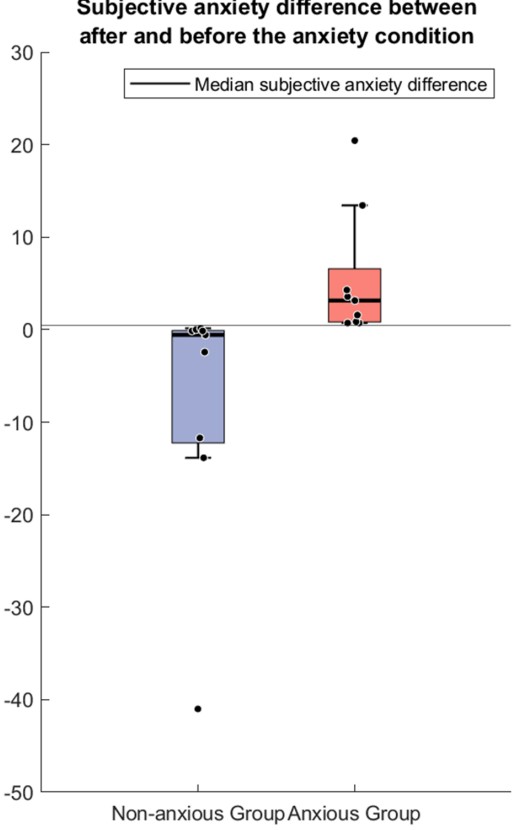

**Fig 2. The differences of the GAD7 questionnaire between after and before the anxiety condition.** This shows the median split for all participants into the non-anxious group and the anxious group.

**Table 1. Statistical information mean, median, standard deviation, 95% confidence interval for each measure for the anxious and non-anxious groups and the rank sum statistics, p-value and cohens'd for the rank sum test between both groups.**

| | Mean anx | Mean non-anx | Median anx | Median non-anx | STD anx | STD non-anx | 95% CI [lower, upper] anx | | 95% CI [lower, upper] non-anx | | Rank sum statistics | P-Value | Cohen's D |
|---|---|---|---|---|---|---|---|---|---|---|---|---|---|
| HR [bpm] | 79.445 | 66.75 | 77.706 | 66.554 | 13.424 | 7.184 | 60.407 | 99.29 | 56.587 | 76.678 | 105 | 0.015661 | 1.1639 |
| HRV [ms] | 0.603 | 0.556 | 0.527 | 0.53 | 0.328 | 0.214 | 0.252 | 1.282 | 0.269 | 0.915 | 85 | 0.67463 | 0.28881 |
| Tonic GSR [μS] | 8.099 | 4.84 | 7.53 | 4.77 | 3.89 | 2.035 | 3.552 | 15.106 | 1.614 | 7.819 | 99 | 0.044392 | 1.0038 |
| Phasic GSR [μS] | 0 | 0 | −0.001 | 0 | 0.019 | 0.012 | −0.03 | 0.034 | −0.019 | 0.019 | 86 | 0.39594 | 0.37244 |
| Pupil size left [mm] | −0.134 | −0.245 | −0.169 | −0.238 | 0.26 | 0.265 | −0.486 | 0.279 | −0.708 | 0.154 | 87 | 0.41968 | 0.50637 |
| Pupil size right [mm] | −0.123 | −0.199 | −0.145 | −0.213 | 0.234 | 0.226 | −0.45 | 0.249 | −0.534 | 0.16 | 88 | 0.56197 | 0.42065 |
| Respiratory Rate [brpm] | 17.879 | 15.37 | 17.816 | 15.902 | 3.229 | 2.622 | 12.64 | 23.273 | 10.392 | 18.767 | 105 | 0.021599 | 1.0127 |
| Head Roll Variance [dps] | 8.261 | 2.104 | 3.39 | 0.721 | 11.954 | 3.278 | 0.34 | 35.031 | 0.099 | 9.584 | 104 | 0.040362 | 1.0291 |

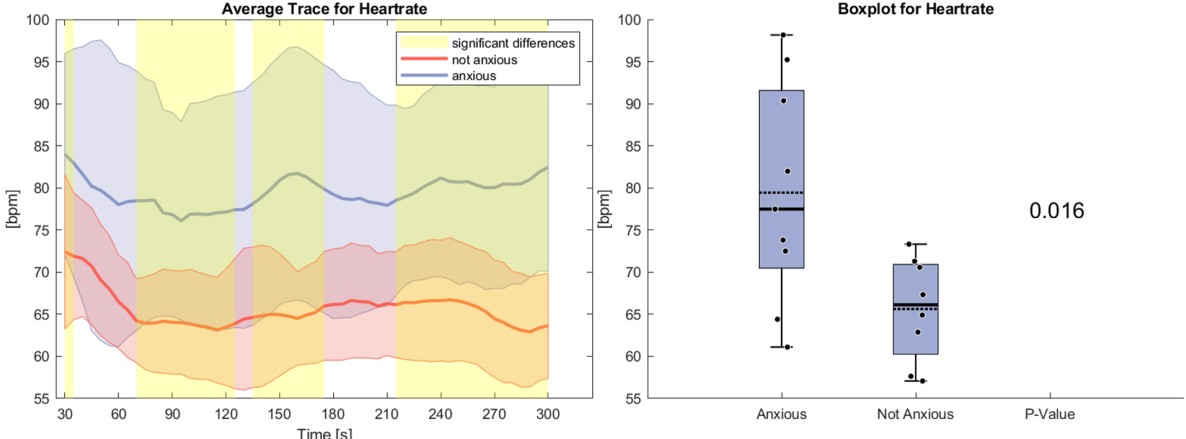

**Fig 3. Left side: Average Trace and standard deviation of heart rate during the anxiety condition.** Right side: Box plot for heart rate averaged over the whole task for the anxious and non-anxious group with a p-value for a Wilcoxon rank-sum test.

the start of the task, both groups exhibit an initial decline, a common effect in scientific studies where participants experience increased tension at the beginning of a new task potentially attributed to the orienting response [33], which is thought to occur when participants experience a change in their environment. While the non-anxious group stabilizes at a consistent level, the anxious group's heart rate gradually rises again, eventually returning to its initial value. The standard deviation is also larger in the anxious group, indicating greater variability. Statistically significant differences are present throughout the task, with the most pronounced effect occurring toward the end as the heart rate of the anxious group increases.

Fig 4 shows the heart rate variability during the anxiety task. Both groups show nearly identical traces and averages over the task, leading to no significant differences. At the beginning of the task we see a decline similar to heart rate.

Fig 5 shows the pupil size of both groups during the anxiety task. The pupil size of the anxious group is generally bigger compared to the non-anxious group but the difference is not large enough to lead to significant differences. While the pupil size declines for both groups over the course of the task, it increases for the last 30 seconds for the anxious group.

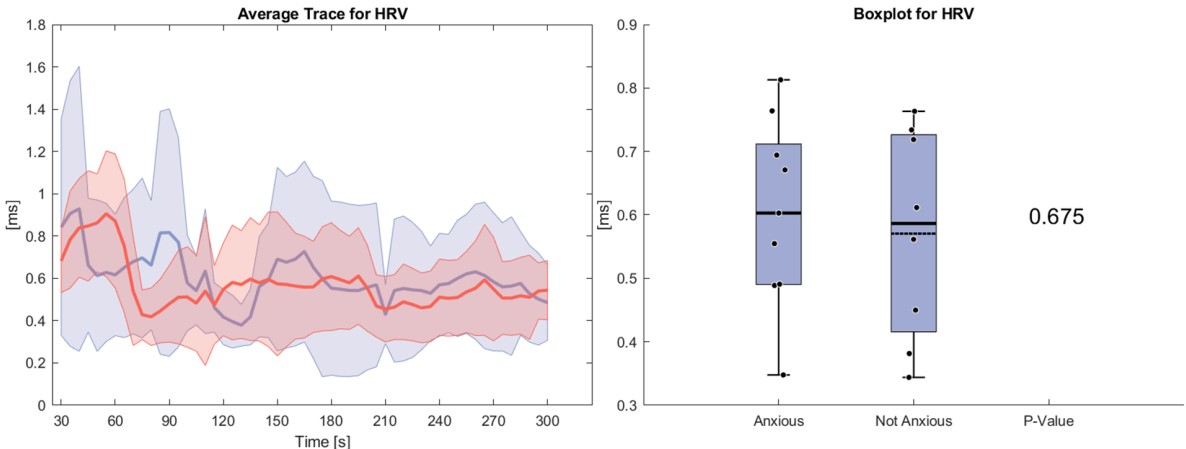

**Fig 4. Left side: Average Trace and standard deviation of heart rate variability during the anxiety condition.** Right side: Box plot for heart rate variability averaged over the whole task for the anxious and non-anxious group with a p-value for a Wilcoxon rank-sum test.

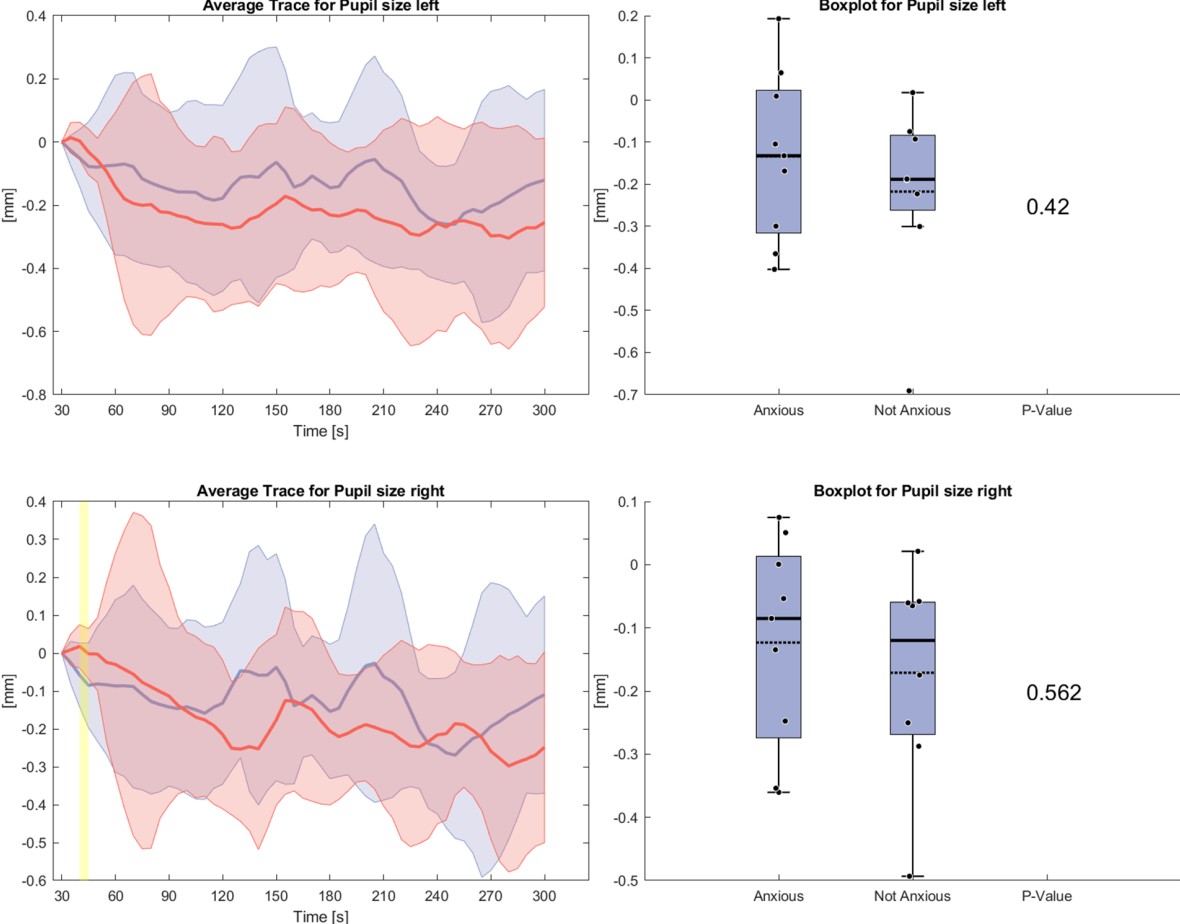

**Fig 5. Left side: Average Trace and standard deviation of the averaged pupil size for the left and right eye during the anxiety condition.** Right side: Box plot for pupil size averaged over the whole task for the anxious and non-anxious group with a p-value for a Wilcoxon rank-sum test.

In Fig 6, the respiratory rate during the anxiety condition is depicted. The anxious group exhibits a significantly higher average respiratory rate, exceeding that of the non-anxious group by 2 brpm. Over time, a divergence emerges: the respiratory rate in the anxious group increases slightly, whereas a slight decrease is observed in the non-anxious group. Unlike heart rate, the variability in both groups remains similar. Statistically significant differences begin to appear after the first 90 seconds.

Fig 7 illustrates the tonic GSR during the anxiety condition, measured in microSiemens (μS). The anxious group shows an average tonic GSR that is significantly higher 3 μS than that of the non-anxious group. Initially, both groups exhibit a drop, similar to what is observed in heart rate. However, while the non-anxious group continues this downward trend, the anxious group gradually reverses it over the course of the task. The standard deviation is greater in the anxious group, again mirroring the pattern seen in heart rate. In this case, statistically significant differences are limited to the first 90 seconds.

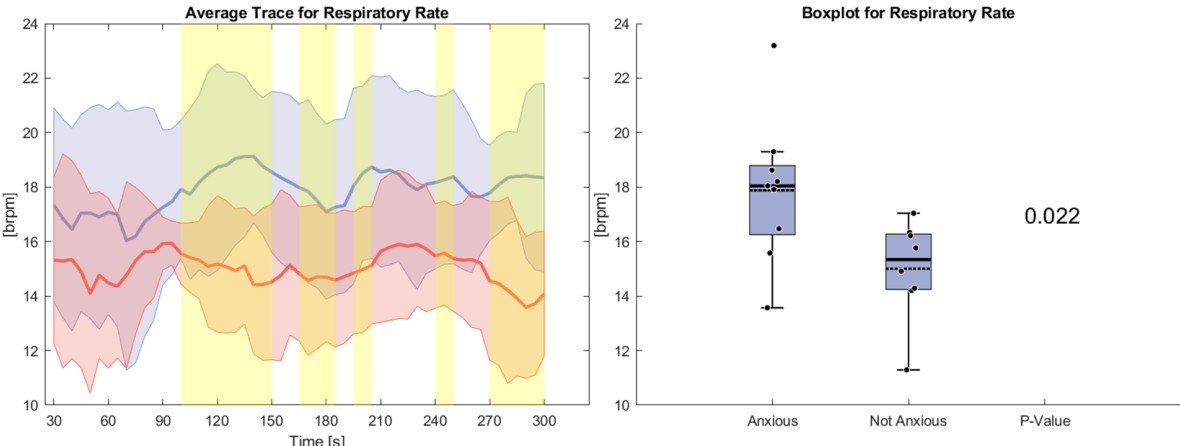

**Fig 6. Left side: Average Trace and standard deviation of respiratory rate during the anxiety condition.** Right side: Box plot for respiratory rate averaged over the whole task for the anxious and non-anxious group with a p-value for a Wilcoxon rank-sum test.

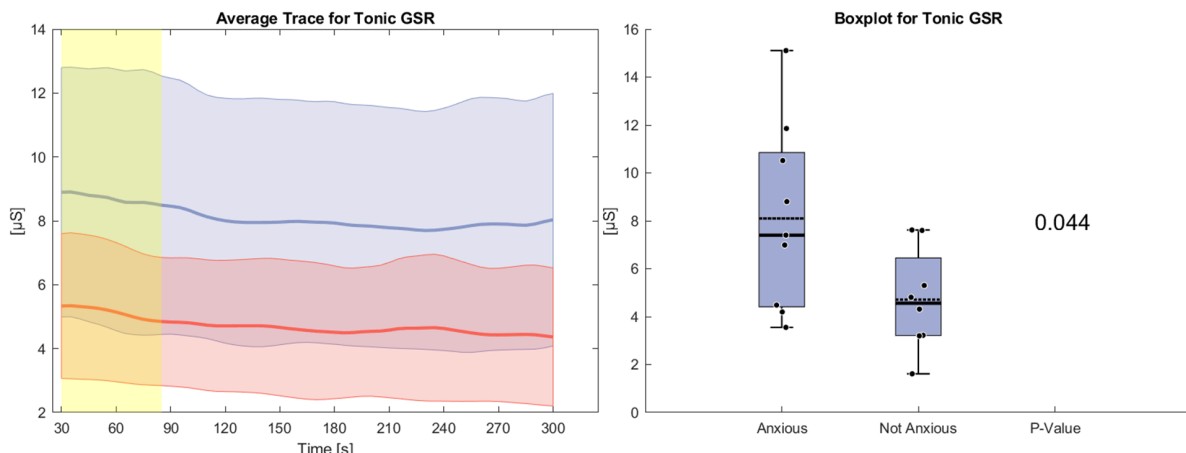

**Fig 7. Left side: Average Trace and standard deviation of tonic GSR during the anxiety condition.** Right side: Box plot for tonic GSR averaged over the whole task for the anxious and non-anxious group with a p-value for a Wilcoxon rank-sum test.

Fig 8 illustrates the phasic GSR during the anxiety condition. Both groups show similar averages of phasic GSR leading to no statistical significance. The anxious group exhibits a large standard deviation compared to the non-anxious group. Standard deviation increases are largest at the beginning and end of the task.

Figs 9 and 10 present the variance and PSD of head rolling motion. The roll variance, measured in degree per second (dps), is generally higher in the anxious group throughout the task similar to heart rate and respiratory rate. Additionally, the standard deviation is greater for anxious participants, with statistically significant differences observed across the task, particularly toward the end. The PSD plot shows uncorrected p-values on the left, corrected p-values in the middle, and significant corrected p-values on the right. Overall, the PSD of head roll motion shows significant differences between the anxious and non-anxious groups in the 5 Hz to 50 Hz range. The Benjamini-Hochberg correction has only a minor effect, with the corrected results closely matching the uncorrected ones.

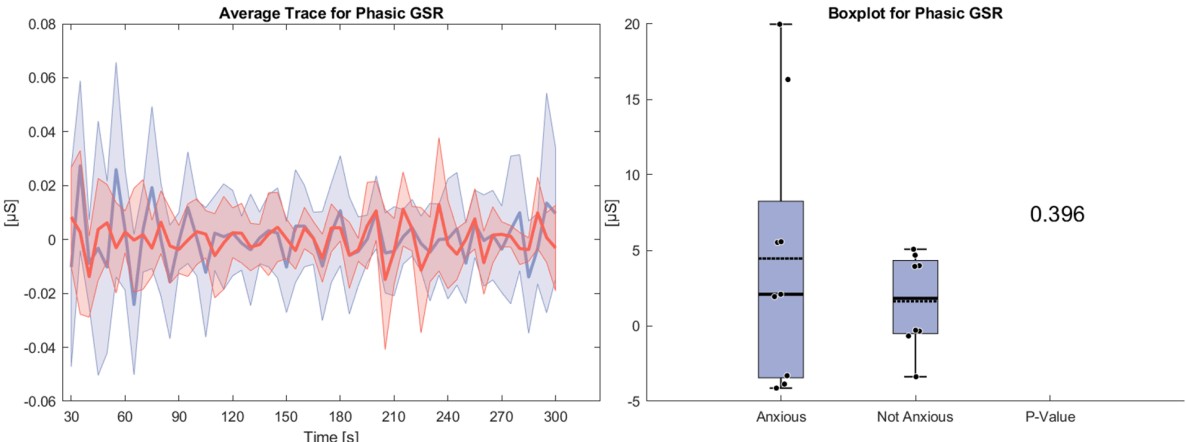

**Fig 8. Left side: Average Trace and standard deviation of phasic GSR during the anxiety condition.** Right side: Box plot for phasic GSR averaged over the whole task for the anxious and non-anxious group with a p-value for a Wilcoxon rank-sum test.

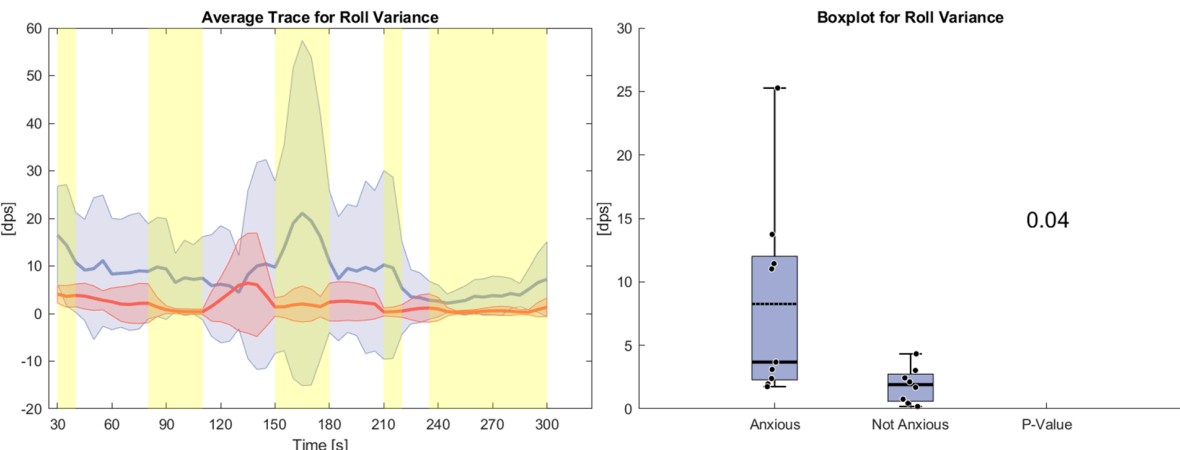

**Fig 9. Left side: Average Trace and standard deviation of Roll Variance during the anxiety condition.** Right side: Box plot for Roll Variance averaged over the whole task for the anxious and non-anxious group with a p-value for a Wilcoxon rank-sum test.

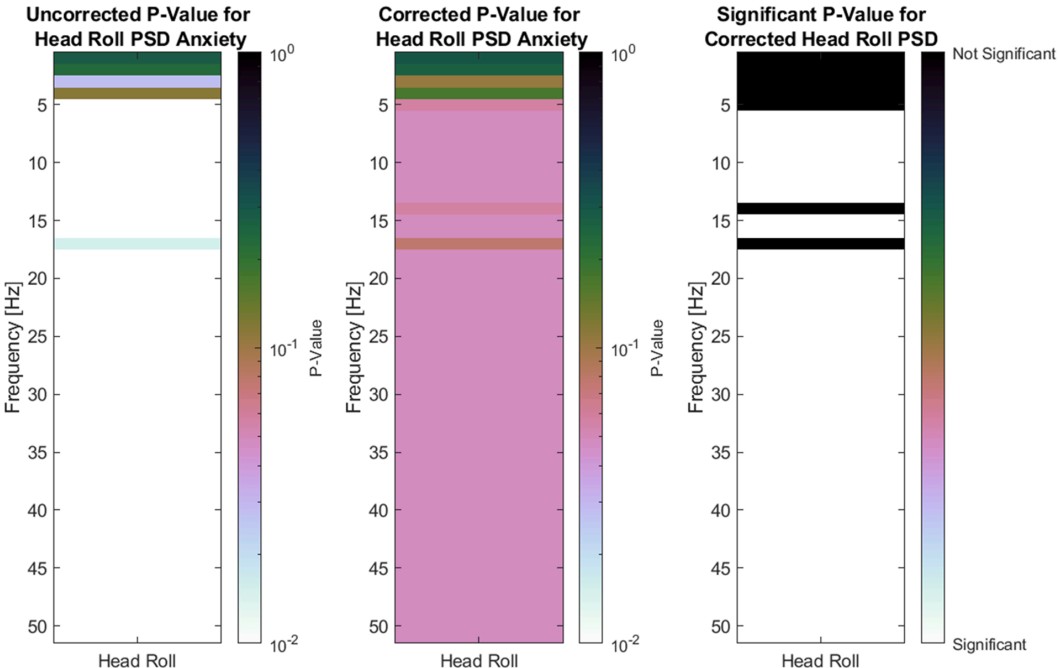

**Fig 10. Left side: Uncorrected P-Values for head roll PSD differences between anxious and non-anxious participants for each frequency band.** Middle: Corrected P-Values for head roll PSD differences between anxious and non-anxious participants for each frequency band. Right side: Binary plot that shows the significance for head roll PSD differences for the anxiety condition.

Fig 11 presents EEG PSD values in three panels: the left panel shows PSD differences, the middle displays corrected p-values, and the right shows uncorrected p-values. In the left panel, PSDs are averaged over the entire task duration and across frequency bands, with medians computed across participants. This method reduces the impact of individual variability and improves overall readability. The middle panel applies the Benjamini-Hochberg correction and shows no statistically significant differences. The right panel presents uncorrected p-values and is included to highlight potential differences that may warrant further investigation. Notable uncorrected differences appear in the theta, beta, and gamma bands. Theta band differences are primarily located in central and left temporal regions, while beta and gamma differences are observed in frontal, temporal, and parietal areas.

## Discussion

We conducted a pilot study to test whether we could detect anticipatory social anxiety through a combination of multiple biosignals. We found that in participants whose anxiety increased during the anticipation period, anxiety was associated with increased heart rate, respiratory rate, tonic GSR, increased variance in head roll, and increases in theta, beta, and gamma power of EEG.

It is noteworthy that we observed a higher absolute heart rate in anxious participants, but no difference in HRV. This finding contrasts with a large body of research linking reduced HRV with increased anxiety [15,16]. However, our participants in this study were all healthy volunteers, and individuals with anxiety disorders were excluded from the study. It is likely that differences in HRV are easier to detect for those with a clinical anxiety disorder.

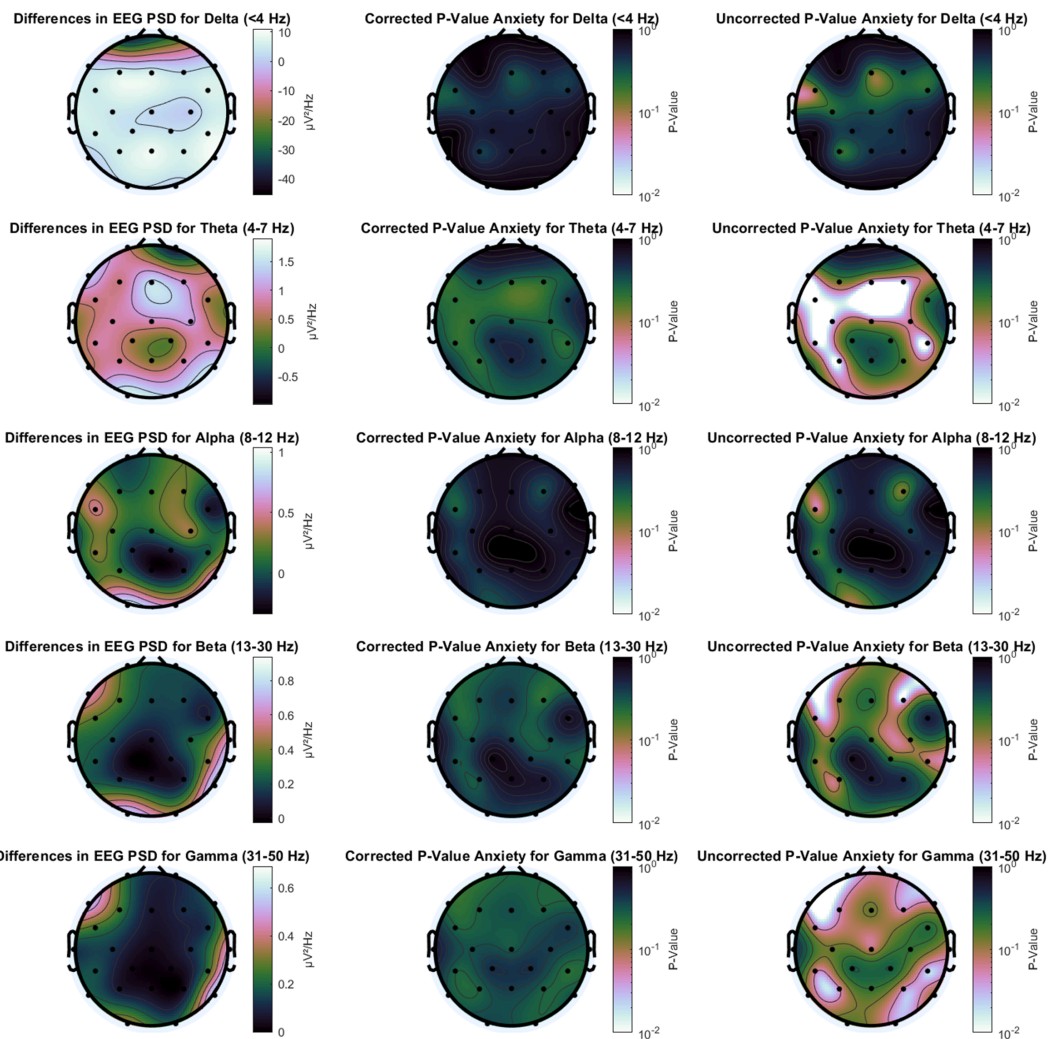

**Fig 11. Left side: Differences in EEG PSD between anxious and non-anxious participants for each EEG frequency band.** Middle: Corrected P-Values for EEG PSD differences for the anxiety condition for each EEG frequency band. Right side: Uncorrected P-Values for EEG PSD differences for the anxiety condition for each EEG frequency band.

Moreover, while decreased HRV is well-established in anxiety disorders, this relationship is less clear for SAD compared to other types of anxiety disorders. A study by Pittig et al. [16], for example, showed that the reduction in heart rate variability (HRV) in individuals with social anxiety disorder (SAD) was weaker compared to other anxiety disorders. Nevertheless, our findings highlight that heart rate should be considered an important factor in psychophysiological profiling of anxiety.

We also found a notable elevated respiratory rate in anxious participants which increased even further the more time progressed. This agrees with similar studies [16,34], where even in baseline conditions participants with SAD had a higher respiratory rate than the control group and where a triggering stimulus increases the rate even further.

The tonic and phasic GSR signals have been shown to be good indicators for stress [21]. It would follow that increased arousal (stress) and subjective anxiety are likely to be associated. From our dataset we could confirm differences for the tonic signal component but not for the

phasic. This is perhaps not surprising as there were no startle events during this task to elicit a phasic response [35]. To our knowledge, there is no published paper directly showing an association between social anxiety and tonic or phasic GSR signals, although stress tests used in the literature often include a social element (e.g. Trier Social Stress Test) [21]. It would be interesting to investigate in further work whether differences in phasic GSR are detectable in those with social anxiety disorder during a naturalistic social interaction.

We found that EEG, specifically PSD features in the theta, beta and gamma power were increased in anxious participants. The significant electrode locations in the beta and gamma bands, found at the front, sides, and back of the scalp, suggest that these features are likely related to movement or muscle tension. Especially above 20 Hz, electromyography activity tends to dominate over EEG [36]. Regarding the increased theta activity, our findings align with previous studies [23], although other research has reported decreased theta power in response to anxiety [37]. The brain can show highly individual responses to stimuli such as workload [38], and the brain's response to increased anxiety is not yet clear. Notably, increased theta power has been observed following unexpected rejection feedback and during social exclusion [22,39], suggesting that theta band activity may play a role in processing social threats. Future research should investigate the specific role of the theta band activity in social threat evaluation.

We did not find a significant difference between groups in pupil size. This is interesting as increased pupil size is associated with noradrenaline release and thus an acute threat response [40]. Similarly with HRV, perhaps we would have seen significantly larger pupil sizes in those with clinical anxiety disorder. Alternatively, perhaps this anticipatory anxiety paradigm does not induce the level of acute threat needed to induce a change in pupil size.

We also examined signals that are not typically studied in anxiety research. The 9-DOF IMU and gyroscope embedded in the eye tracker detected a significant difference in one specific type of movement: the rolling motion of the head. The power of this movement across multiple frequency bands above 5 Hz, as well as its variance, showed notable differences in participants with higher anxiety levels, which were attributed to distinct movements or muscle tension. While some studies have analyzed head movements during anxiety-inducing tasks [41,42], their findings have been inconclusive, partly because head movements are also influenced by stimuli. Since our study focused on anticipatory anxiety without external stimuli, our results may more accurately reflect head movement responses. These results show that recording a range of signals could uncover new links between physiological responses and anxiety.

We believe our approach of combining multiple biomedical measures to be highly novel. However, while significant differences were found in several biosignal measures between groups of participants in this pilot, individual responses often vary, particularly in EEG data. Further work should explore inter-individual differences in biosignal responses as well as group-level findings. Such analyses were limited in the present pilot by our sample size. A larger study could employ cluster or factor analysis to identify subgroups within the participants and correlate them with specific symptoms for more specific phenotyping.

There are some potential limitations of our pilot study. First, the anxiety condition was anticipatory in nature, and perhaps we would have seen changes to the physiological profile during an acute threat condition, such as an actual social interaction. Due to this being a pilot study, we only tested a small number of participants. Nevertheless, we saw significant differences within this small sample, highlighting the potential size of the effects on biosignal measures. However, as this was a pilot study assessing feasibility of our approach, we were unable as yet to conduct analyses on combinations or profiles of signals, which is our ultimate aim. Therefore, our results should be interpreted accordingly. Finally, all participants were healthy

volunteers. It is presently unknown whether we would find the same profile in those diagnosed with social anxiety disorder. If the profile differs in patients, then these variables could potentially be used diagnostically and/or for prognostication.

## Conclusions

New, more effective therapies are needed for SAD. In this pilot study we showed that anticipatory social anxiety can trigger changes in a range of biomedical signals namely, respiratory rate, heart rate, tonic GSR, head roll, and central EEG theta activity. We suggest that these signals should be included in montages when studying anxiety and exploring potential options for treatment development. Future research should identify if additional signals not tested here, such as electromyography, should also be included. Furthermore, future work should assess whether the profile changes during acute threat (fear/startle) and whether patients with clinical social anxiety disorder exhibit a differing profile.

## Supporting information

**S1 File. Complete experimental protocol.**
(PDF)

## Author contributions

**Conceptualization:** Christoph Tremmel, Nathan T. M. Huneke, M. C. Schraefel.

**Data curation:** Christoph Tremmel.

**Formal analysis:** Christoph Tremmel.

**Funding acquisition:** Christoph Tremmel, Nathan T. M. Huneke, M. C. Schraefel.

**Investigation:** Christoph Tremmel, Daniel Hobson.

**Methodology:** Christoph Tremmel, Nathan T. M. Huneke, M. C. Schraefel.

**Project administration:** Christoph Tremmel.

**Resources:** Christoph Tremmel.

**Software:** Christoph Tremmel.

**Supervision:** Christoph Tremmel, M. C. Schraefel.

**Validation:** Christoph Tremmel.

**Visualization:** Christoph Tremmel.

**Writing – original draft:** Christoph Tremmel, Nathan T. M. Huneke, M. C. Schraefel.

**Writing – review & editing:** Christoph Tremmel, Nathan T. M. Huneke, Daniel Hobson, M. C. Schraefel.

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
