## [Decision Letter · Decision Letter 0]

16 Jun 2025

PONE-D-25-14608Detection of Social Anxiety Using Multiple Simultaneous Biosignals: A Pilot StudyPLOS ONE

Dear Dr. Tremmel,

Thank you for submitting your manuscript to PLOS ONE. After careful consideration, we feel that it has merit but does not fully meet PLOS ONE’s publication criteria as it currently stands. Therefore, we invite you to submit a revised version of the manuscript that addresses the points raised during the review process.

**ACADEMIC EDITOR:** - please do follow very constructive comments of our reviewers 

We look forward to receiving your revised manuscript.

Kind regards,

Prof. Dr. Dragan Hrncic, MD, MSc, MBE, PhD

Academic Editor

PLOS ONE

“The authors acknowledge the University of Southampton's Researcher Development Concordat team for funding consumables in support of this study;  the Web Science Institute Southampton that provided the funds for the experiment and equipment as part of a interdisciplinary pilot project scheme. This research is also supported by UKRI/EPSRC Grants EP/T007656/1: Health Resilience Interactive Technologies and EP/R029563/1: AutoTrust.”

Reviewers' comments:

Reviewer's Responses to Questions

**Comments to the Author**

1. Is the manuscript technically sound, and do the data support the conclusions?

Reviewer #1: No

2. Has the statistical analysis been performed appropriately and rigorously? 

Reviewer #1: No

3. Have the authors made all data underlying the findings in their manuscript fully available?

Reviewer #1: No

4. Is the manuscript presented in an intelligible fashion and written in standard English?

Reviewer #1: Yes

5. Review Comments to the Author

Reviewer #1: Authors present a compelling argument. In their study they tested, in a small sample of university students and staff, on a number of biosignals whether they can be used to differentiate between low and high anxiety groups during an anticipatory anxiety task. Their findings indicate that some signals showed a significant difference between the two groups. This is an interesting dataset and research quest worth exploring, however, in the current stage, I see minor improvements the manuscript needs before it will be ready for publishing. That is, in summary, a clearer presentation of methods, results, and conclusion. Introduction develops well the understanding of mechanisms of SAD. Discussion is satisfactory, and authors are aware of limitations of their study.

Major points

1. In abstract, manuscript states to test whether anticipatory anxiety can be identified through a combination of signals, on line 39 a 'profile' of features together is to be explored. However, no analysis on combinations or profiles is presented, and no conclusions can be drawn to that end. Here, each signal was tested separately, and limited conclusions can be drawn in current design whether that is due a single latent factor such as vigilance, or due to a specific set of latent variables that may or may not make up a phenotype of SAD. Language should be corrected accordingly to available evidence.

2. Methods section is missing part on statistical analysis. Steps to statistical analysis should be explained, and how the problem of family wise error rate was dealt with.

3. Given the small sample and heterogenity of data, descriptive and inferential statistics should be included. Statements that one group had higher score than the other need to be supported by mean/median, STD/IQR, t / F value, and p value. If p value corrections seem too conservative, a permutation approach may help, and authors may present a p value associated with a position of the original t/F value in the distribution of permuted t/F values.

Minor points requiring clarification

1. Intro to abstract reflects need for effective therapies, but not for what the study is doing

2. Sample is not described in abstract

3. Conclusion is not linked to abstract introduction

4. Line 8 unclear how cost is related to SAD

5. Line 23 - personal believes should be based on preliminary evidence or case reports

Introduction also explains well that a single marker may be overgeneralizing. This may be an issue with results of this study, too.

6. Line 42, accronym EDA is not explained

7. Line 72, experimental procedure, high-pass analog filter (on recording device) should be included, especially when interpreting very slow oscillations

8. From the description of experimental task, in the supplementary material, or original paper by Huneke I cannot figure when was the GAD administered. Was it before the task was introduced? and Before it was revealed that experiment is not going to include interview, or after that? I would appreciate clarification on this to understand the results.

9. Line 109 - Could you elaborate on how did you split the group? I can clearly see that some people in anxious groups have lower values on GAD than in non-anxious in Fig2. This should be in the part Statistical analysis. This part may also include what significance test was used and why, and how was treated the multiple comparisons problem

10. line 160 - could add 'Head roll' for clarity

11. Line 162 - signal filtered 0.5 - 50Hz, but downsampled to 100Hz, while this satisfies nyquist frequency, higher frequencies should be interpreted cautiously, as caution commands that frequencies higher than 1/3 or 1/4th of sampling frequency should not be analysed.

12. Lines 169-170. Claims higher or lower should be accompanied by descriptive statistics and inferential statistics

13. Fig 2 description may for clarity include how groups were split to be stand-alone understandable

14. Lines 175 to 179 belong to methods

15. Fig 10. Head rolls need more explanation. Head motion is traditionally measured as oscillation in eg parkinsons, or head boobing and sway is measured below 5Hz. Oscillations of head are not usually present, and do not appear meaningful. This result is intriguing, but without further context, and without correction for multiple comparisons, may be spurious. Perhaps a histogram or boxplot of values averaged over significant frequencies would provide further credibility?

16. Line 247 - 'marginally significant' is not meaningful. Size of an effect is not measured by a p value.

17. Line 277 - If it was not significant, there was no difference. Sometimes, in pilot data, we can speak about 'trends'. However, fig 5. right appears to show homogenous groups and one extreme value.

Kind regards,

Marek Nikolic

6. PLOS authors have the option to publish the peer review history of their article (what does this mean?). If published, this will include your full peer review and any attached files.

Reviewer #1: **Yes: **Marek Nikolič

---

## [Author Response · Author response to Decision Letter 1]

18 Jul 2025

All the responses to the reviewer are in the pdf document: "Letter to the reviewer Detection of Social Anxiety.pdf"

---

## [Editor Report · Decision Letter 1]

5 Aug 2025

Detection of Social Anxiety Using Multiple Simultaneous Biosignals: A Pilot Study

PONE-D-25-14608R1

Dear Dr. Tremmel,

We’re pleased to inform you that your manuscript has been judged scientifically suitable for publication and will be formally accepted for publication once it meets all outstanding technical requirements.

Kind regards,

Prof. Dr. Dragan Hrncic, MD, PhD 

Academic Editor

PLOS ONE
---

## [Editor Report · Acceptance letter]

PONE-D-25-14608R1

PLOS ONE

Dear Dr. Tremmel,

I'm pleased to inform you that your manuscript has been deemed suitable for publication in PLOS ONE. Congratulations! Your manuscript is now being handed over to our production team.

Kind regards,

on behalf of

Professor Dragan Hrncic

Academic Editor

PLOS ONE